# Substrate Stiffness Modulates Renal Progenitor Cell Properties via a ROCK-Mediated Mechanotransduction Mechanism

**DOI:** 10.3390/cells8121561

**Published:** 2019-12-03

**Authors:** Maria Elena Melica, Gilda La Regina, Matteo Parri, Anna Julie Peired, Paola Romagnani, Laura Lasagni

**Affiliations:** 1Centre for Research, Transfer and High Education for the development of DE NOVO Therapies (DENOTHE), Viale Morgagni 50, 50136 Florence, Italy; mariaelena.melica@gmail.com (M.E.M.); anna_peired@hotmail.com (A.J.P.); paola.romagnani@unifi.it (P.R.); 2Department of Clinical and Experimental Biomedical Sciences “Mario Serio”, University of Florence, Viale Morgagni 50, 50134 Florence, Italy; gildalaregina@hotmail.it (G.L.R.); matteo.parri@unifi.it (M.P.); 3Nephrology Unit and Meyer Children’s University Hospital, Viale Pieraccini 24, 50139 Florence, Italy

**Keywords:** renal progenitor cells, Young’ modulus, tissue engineering, regenerative medicine, Rho-kinase

## Abstract

Stem cell (SC)-based tissue engineering and regenerative medicine (RM) approaches may provide alternative therapeutic strategies for the rising number of patients suffering from chronic kidney disease. Embryonic SCs and inducible pluripotent SCs are the most frequently used cell types, but autologous patient-derived renal SCs, such as human CD133+CD24+ renal progenitor cells (RPCs), represent a preferable option. RPCs are of interest also for the RM approaches based on the pharmacological encouragement of in situ regeneration by endogenous SCs. An understanding of the biochemical and biophysical factors that influence RPC behavior is essential for improving their applicability. We investigated how the mechanical properties of the substrate modulate RPC behavior in vitro. We employed collagen I-coated hydrogels with variable stiffness to modulate the mechanical environment of RPCs and found that their morphology, proliferation, migration, and differentiation toward the podocyte lineage were highly dependent on mechanical stiffness. Indeed, a stiff matrix induced cell spreading and focal adhesion assembly trough a Rho kinase (ROCK)-mediated mechanism. Similarly, the proliferative and migratory capacity of RPCs increased as stiffness increased and ROCK inhibition, by either Y27632 or antisense LNA-GapmeRs, abolished these effects. The acquisition of podocyte markers was also modulated, in a narrow range, by the elastic modulus and involved ROCK activity. Our findings may aid in 1) the optimization of RPC culture conditions to favor cell expansion or to induce efficient differentiation with important implication for RPC bioprocessing, and in 2) understanding how alterations of the physical properties of the renal tissue associated with diseases could influenced the regenerative response of RPCs.

## 1. Introduction

Kidney disease is a worldwide public health problem [1]. The number of patients with acute and chronic renal failure worldwide, which can progress to end-stage renal disease (ESRD) with high mortality rate, is increasing thus creating the need for clinicians and investigators to search out alternative treatments other than dialysis and/or kidney transplantation. Stem cell (SC)-based therapies—tissue engineering (TE) and regenerative medicine (RM)—may provide attractive approaches [2]. TE strategies combine supporting scaffolds, of either natural or biodegradable polymers, or decellularized kidneys with different kinds of SCs and growth factors, to generate organoids, implantable renal assist devices, or transplantable organs [2,3,4,5]. On the other hand, RM approaches rely on the direct administration of different types of SCs, such as embryonic SCs (ESCs) [6,7,8], inducible pluripotent SCs (iPSCs), fetal SCs (FSCs), amniotic fluid-derived SCs [9], or adult SCs [10], to aid renal regeneration in the context of both acute kidney injury and chronic kidney disease [11]. In both cases, demand is increasing for alternative cell sources that overcome the ethical and legal issues accompanying the application of ESCs and FSCs, devoid of the mutational effects associated with iPSCs, and of autologous origin.

Encouraging the intrinsic regenerative capacity of the kidney through pharmacological treatments that target populations of renal progenitor cells (RPCs) residing inside the kidney is an alternative RM approach [12]. Indeed, recent data obtained in mouse models of kidney disease demonstrated that pharmacological enhancement of glomerular and tubular regeneration driven by a population of Pax2+ RPCs is feasible and effective [13,14,15].

The Pax2+ cells are the murine counterparts of a population of RPCs described and characterized in humans as a hierarchical population of CD133+CD24+ cells, containing parietal epithelial cells and scattered tubular epithelial cells [16,17,18]. Human RPCs exhibit cellular plasticity and stem-like properties, such as multipotency, that is the capability to differentiate in vitro into several types of renal epithelial cells, including podocytes and tubular epithelial cells [17,18,19]. Additionally, we demonstrated the possibility to retrieve autologous cells from the patient’s urine [20]. Thus, RPCs represent an attractive, but still unexplored, alternative cell source for TE applications and a promising target to foster kidney repair in situ.

Both RPC-based TE and RM approaches would greatly benefit from advances in our understanding of the factors that influence their behavior. Indeed, it is clear that inherent factors, present in the environment of the cell or in cell culture, are critical to maintaining their undifferentiated state, supporting specific RPC fate decisions or shift in phenotype. Recent developments emphasized that physical attributes of the SC niche and of the materials in the cell environment can influence SC behavior with a potency that rivals that of biochemical supplements [21,22]. In particular, the mechanical elasticity of the substrate, or stiffness, is a physical property defined as force per unit area divided by deformation (Young’s modulus, E) that greatly affects cell activity in vitro [23,24,25] and in vivo, during development, in tissue homeostasis, and in disease progression [26]. In the kidney, the mechanical environment is subjected to modifications in established models of glomerular diseases [27] and can affect the differentiated state of numerous cell types, including podocytes [28,29,30,31].

In this paper, we explore the impact of substrate stiffness on RPC viability, proliferation, and differentiation and try to identify the mechanisms that enable them to “feel” the stiffness and convert biomechanical stimuli into biochemical signals. We found that the morphology, proliferation and migration of RPCs were highly dependent on mechanical stiffness. Indeed, stiff matrix induced cell spreading and focal adhesion (FA) assembly trough a Rho kinase (ROCK)-mediated mechanism. Similarly, the proliferative and migratory capacity of the cells increased as stiffness increased and ROCK inhibition by antisense LNA-GapmeRs or Y27632 abolished these effects. The capacity of RPCs to differentiate into the podocyte lineage, in terms of acquisition of podocyte markers, was also modulated in a narrow range, by the elastic modulus of the substrate and involved ROCK activity.

These results would help (1) to set-up new optimized RPC culture conditions to favor cell expansion or induce efficient differentiation with important implication for RPC bioprocessing in view of their application in TE; (2) to understand how the regenerative response of RPCs could be influenced by alterations of the physical properties of the renal tissue associated with diseases.

## 2. Materials and Methods

### 2.1. RPC Cultures

Human RPCs were obtained and cultured as previously described [16] in agreement with the Ethical Committee on human experimentation of the Azienda Ospedaliero-Universitaria Careggi, Florence, Italy (All subjects gave their informed consent for inclusion before they participated in the study. The study was conducted in accordance with the Declaration of Helsinki, and the protocol was approved by the Ethics Committee on Human Experimentation of the Azienda Ospedaliero-Universitaria Careggi, Florence, Italy (Project identification code 2015/0009082 from 25/03/2015)). Primary cultures were checked for CD133 and CD24 expression by FACS analysis, as described [16]. Primary cultures with a percentage of CD133+CD24+ cells higher than 95% were used at passage 2–5. Softwell G 96, Collagen coated, Petrisoft 35 collagen coated, and Softslip collagen coated plates, were from Matrigen (Cell Guidance Systems Ltd., Cambridge, UK). Cell proliferation was assessed by MTT assay (Thermo Fisher Scientific, Waltham, MA, USA) following manufacturer’s instructions. For the generation of growth curves, 10,000 RPCs were plated in EGM-MV (Lonza, Basel, Switzerland) + 20% FBS (Hyclone, GE Healthcare Life Sciences, Logan, UT, USA) on Petrisoft. After 1 or 4 days, cells were detached with trypsin-EDTA solution 0.25% (Sigma Aldrich, St. Louis, MO, USA) and counted.

For Rho-kinase inhibition, RPCs were seeded on the plate and, after 12 h, treated with vehicle (DMSO) or Y27632 (10 µM) (Merck KGaA, Darmstadt, Germany).

### 2.2. ROCK Inhibition with Antisense LNA-GapmeRs

ROCK1 and ROCK2 antisense LNA-GapmeRs were designed by the Qiagen selection tool (Qiagen, Hilden, Germany) and administered to RPCs using Xfect transfection reagent (Takara Bio, Kusatsu, Japan). In detail, 16,000 cells/well were seeded in a 24 well plate and transfected overnight with 1 µM antisense LNA-GapmeRs for ROCK1, ROCK2 or Negative control (scramble). The efficacy of mRNA knockdown was evaluated 48 h after transfection by qRT-PCR. For cell proliferation assessment, we transfected RPCs cultured on the Softwell G 96-Collagen coated plate and evaluated MTT after 48 h. Similarly, immunofluorescence staining for ROCK1 and ROCK2 was performed 48 h after transfection performed on RPCs cultured on Softslip.

### 2.3. Cell Viability Assay

Cell viability assay was performed employing Calcein-AM (Invitrogen, Carlsbad, CA, USA) staining following manufacturer’s instructions.

### 2.4. RPC Shape Descriptors

For analysis of RPC circularity, aspect ratio and spreading area, RPCs were cultured on Softwell of different stiffness for 7 days. At least 40 cells for each condition were analyzed using ImageJ. Cell spreading and circularity were measured manually tracing the outline of phase images and quantified using ImageJ. Major to minor axis ratio was analyzed using ImageJ. At least 70 cells were analyzed per condition.

### 2.5. RPC In Vitro Differentiation

Podocyte differentiation was obtained as previously described [18], by culturing the cells for 48 h in DMEM/F12 medium (Sigma Aldrich) + 10% fetal bovine serum FBS (Hyclone,) and 100 µM all-trans retinoic acid (RA, Sigma-Aldrich) in presence or absence of Y27632 (10 µM).

### 2.6. qRT-PCR

Quantitative RT-PCR was performed as described [20] using commercially available Assay on Demand kits (Applied Biosystems, Warrington, UK).

### 2.7. Immunofluorescence and Confocal Microscopy

Confocal microscopy was performed on RPCs cultured on Softslip by using a Leica SP5 AOBS confocal microscope (Leica Microsystems, Wetzlar, Germany) equipped with a Chameleon Ultra-II two-photon laser (Coherent, Milan, Italy). Cells were fixed for 10 min with paraformaldehyde. The following antibodies were used: Anti-paxillin (ab32084, dilution 1:250, Abcam, Cambridge, UK), anti-nephrin (NMP1-30130Y17-R, dilution 1:200, Novus Biologicals Centennial, CO, USA), anti-ROCK1 (Cat# 202694T02, dilution 1:200, Sino Biological, Wayne, PA, USA), anti-ROCK2 (Cat# HPA044109, dilution 1:30, Sigma Aldrich, Milan, Italy). Staining with Alexa Fluor 546 phalloidin (Cat# A22283, dilution 1:100, Life Technologies, Monza, Italy) was performed following manufacturer’s instructions. Double immunolabelling was performed as described [13] and Alexa-Fluor secondary antibodies were from Molecular Probes (Life Technologies). Nuclei were counterstained with DAPI (Life Technologies) and excited with multiphoton laser at 800 nm.

### 2.8. Image Analysis

The intensity of phalloidin and nephrin stainings was quantified with ImageJ by measuring raw integrated densities of the signal. Background raw integrated densities were subtracted, and this net integrated density was then normalized to the total measured cell area. At least 10–15 cells were analyzed per condition in two independent experiments.

### 2.9. Live Cells Imaging and Random Motility Analysis

RPCs were seeded on the Softslip and incubated 24 h to allow cells to fully spread. The Softslip was then transferred to the microscopy facility and time-lapse recordings were performed using Leica SP5 microscope equipped with a CO2 incubator at 37 °C. Microphotographs of a defined region were taken with a 10× objective at 5 min intervals for 9 h to generate a video. For random motility analysis, at least 20 cells for each conditions were manually tracked for at least 3 h using the ImageJ MtrackJ plugin. Data were used to calculate cell trajectories, average cell speed, and mean square displacement using the DiPer open source program [32].

### 2.10. Statistical Analysis

The results were expressed as mean ± SEM or as median (interquartile range). Comparison between groups was performed by the one-way ANOVA with Tukey or Bonferroni post-hoc test, as appropriate. Statistical analysis was performed using OriginPro statistical software. A *p*-value < 0.05 was considered statistically significant.

## 3. Results

### 3.1. Substrate Stiffness Controls RPC Phenotype and Proliferative Capacity In Vitro

The shape of individual cells is based on the balance between external biomechanical forces and internal cellular forces, and the level of internal forces is proportional to the elastic material properties of the surrounding extracellular matrix (ECM) [33]. This suggests that cell shape can be controlled through both matrix elasticity and biomechanical forces. It was therefore hypothesized that specific RPC shapes could be generated as a function of the stiffness of the substrate. To investigate the response of RPCs to different levels of stiffness, we cultured them on collagen I-coated polyacrylamide hydrogels with variable stiffness, ranging from 0.2 kPa to 50 kPa, grafted onto standard cell culture formats. Following 48 h of culture, RPC morphology was evaluated through contrast phase microscopy (Figure 1a–c). RPCs cultured on substrates of low stiffness (0.2–2 kPa) were small and presented long protrusions that extended from the plasma membrane reaching other cells over long distances (over 100 µm), similar to tunneling nanotubes (Figure 1a). Moreover, during culture on soft substrate, RPCs organized in clusters giving rise to cord-like structures (Figure 1b). On substrates of higher stiffness (from 4 to 50 kPa), RPCs grew as isolated cells exhibiting thin, sheet-like membrane protrusions resembling lamellipodia (Figure 1c). To quantitatively assess RPC morphology, we measured shape descriptors such as circularity, aspect ratio and cell spread using phase-contrast microscopy and image processing with the computational tool ImageJ. We observed that RPCs seeded on the hydrogels exhibited stiffness-dependent circularity, aspect ratio and cell spreading area (Figure 1d–f and Appendix A). In particular, circularity and cell spreading area remained almost unchanged from 0.2 to 2 kPa, significantly increased at 4 kPa, and remained then stable up to 50 kPa (Figure 1d,e and Appendix A). On the contrary, major to minor axis ratio decreased proportionally to the stiffness, meaning that RPCs acquired more symmetric shapes on higher stiffness substrates (Figure 1f and Appendix A). Thus, our results suggest that RPCs respond to the stiffness of the substrate by changing their morphology and shape.

The shape of cells is a fundamental signal of proliferation, a potent regulator of cell growth and physiology, and is adapted for specific functions [34,35]. We thus investigated the effects of different substrate stiffness on the proliferative capacity of RPCs. Based on the results obtained above, we narrowed down the analysis to five representative Young’s modulus, i.e., 0.5, 2, 4, 12, and 50 kPa. Using the MTT assay, we observed that the proliferation rate of RPCs increased proportionally to the substrate stiffness, as showed in Figure 2a. However, the MTT readout is a measure of total metabolic activity in a cell culture and is a mixture of cell proliferation rate, metabolic rate and cell survival. Thus, to verify whether the increase detected in the MTT assay corresponded to an increase in cell number, we seeded an equal number of RPCs on different substrates and counted them after one and four days in culture. In accordance with the MTT results, we observed an increase in cell number proportional to the stiffness of the substrate (Figure 2b). Interestingly, RPCs seeded on soft substrates (0.5 kPa) exhibited a hampered capacity to adhere to the substrate in comparison to the 50 kPa stiffness, as demonstrated by the higher number of cells present in the surnatant one day after plating (Figure 2c). Since RPCs are anchorage-dependent cells, they will die by anoikis following detachment from the ECM. However, the difference in the percentage of non-adherent cells on 0.5 kPa with respect to the 50 kPa substrate (16% vs 7%) was too small to account for the high difference in cell number observed at day 4 between these two conditions. Moreover, while no difference in the number of initial seeded cells was observed between the 2 and the 12 kPa stiffness substrates at day 1, a significant difference was observed at 4 days, that therefore could be attributed to quiescence of RPCs seeded on soft substrate (0.5 and 2 kPa) and to proliferation on stiffer substrate. Calcein-AM/propidium iodide staining was also performed to verify the effect of the substrates on RPC viability. As showed in Figure 2d, the mechanical property of the substrate did not alter RPC viability, as demonstrated by the absence of propidium iodide positive cells in all conditions. On the contrary, we could confirm that the stiffness of the substrate strongly influenced RPC proliferation, as demonstrated by the higher number of cells present after four days of culture on plates with higher Young’s modulus (Figure 2d). Altogether, these results demonstrated that substrate stiffness modulates the RPC proliferative capacity.

### 3.2. Substrate Stiffness Modulates Cytoskeleton Organization and FA Formation

Cytoskeleton organization and FA formation are notoriously involved in converting mechanical cues into intracellular signals [36,37,38], thus regulating cell shape [38,39] and downstream cellular activities, e.g., migration [39] and proliferation [40]. Paxillin is a major component of FA complexes, and its clustering is characteristic of the formation of FA [41]. Therefore, organization of cytoskeletal F-actin and the presence of paxillin patches within RPCs cultured on substrate with different stiffness were analyzed by immunofluorescence using confocal microscopy (Figure 3a,b). RPCs on 0.5 and 2 kPa hydrogel showed a decreased spreading area with a rigidity-dependent dissipation of stress fibers (Figure 3a,b). In contrast, RPCs cultured on stiff substrates (4–50 kPa) were typically well-spread with brighter F-actin displaying a bundle-like distribution (actin stress fibers) (Figure 3a,b). In RPCs grown on soft hydrogel substrates, paxillin expression was low and with diffuse distribution (Figure 3a,b), while the percentage of cells presenting paxillin distributed in intense clusters localized specifically at the end of bundle-like actin microfilament, and the number of paxillin patches per cell increased in a stiff-dependent manner (Figure 3c,d).

These results showed a strong correlation between the mechanical properties of the substrate and actin cytoskeleton reorganization and FA assembly in RPCs.

### 3.3. Substrate Stiffness Modulates RPC Migration In Vitro

To assess the effect of substrate stiffness on RPC motility, we monitored cells in real time using time-lapse microscopy and analyzed cell movement through the open-source computer program DiPer [32]. Following tracking, we analyzed cell trajectories, cell speed and mean square displacement (MSD). Figure 4a–e shows representative wind-rose plots of cell trajectories on 0.5, 2, 4, 12, and 50 kPa, demonstrating the difference in cell migration capacity of RPCs grown on substrates with different E. In particular, we could demonstrate that RPC migration was limited on the 0.5 and 2 kPa stiffness, increased on the 4 kPa substrate and remained stable on the higher stiffness plates. Similarly, cell speed, defined as the average of all instantaneous speed for all cells, was higher on substrates of 4, 12, and 50 kPa with respect to that observed on the soft substrates (Figure 4f). In the context of cell migration, MSD is a good measure of the surface area explored by cells over time, which relates to the overall efficiency of migration. MSD increased proportionally to the stiffness of the substrate (Figure 4g).

These results suggest that RPC migration rate is substrate-stiffness sensitive, with the 4 kPa elastic modulus being the cutoff for RPCs to shift from a low to a high migratory behavior.

### 3.4. Substrate Stiffness Modulates ECM Production and RPC Differentiation In Vitro

We already demonstrated that RPCs produce ECM components in response to transforming growth factor-β (TGF-β) [42]. We thus investigated whether this capacity could be modulated by substrate stiffness by evaluating the mRNA expression of collagen Iα1 (COL1A1), collagen IVα1 (COL4A1), fibronectin (FN1) and laminin (LAMB2) following 2 days of culture in presence or absence of 10 ng/mL of TGF-β. Interestingly, we observed that the production of ECM was inversely related to the Young’s modulus of the substrate (Figure 5a), independently from the presence of TGF-β (Figure 5b).

Direct effects of matrix physical attributes such as matrix stiffness on SC lineage specification have been demonstrated [25]. It is also known that responses of cells to soluble inducers, such as growth factors, couple to matrix anchorage [25]. We therefore verified whether substrate stiffness could elicits a pro-differentation effect of RPCs though mechanotransduction. RPCs were cultured for 2 weeks on substrates with different stiffness. Then, mRNA was extracted and markers of stemness quantified. Expression of Pax2, a progenitor cell-related transcription factor, and Oct4, a stem cell marker, by qRT-PCR analysis showed a trend of increased expression in RPCs plated on soft substrates when compared to stiffer substrates (Figure 5c), in line with what was already described for mesenchymal stem cells (MSCs) and fibroblasts [43,44], thus suggesting that short-term culture on a soft environment maintains RPCs in a more undifferentiated state.

We then assessed the effect of substrate stiffness on the capacity of RPCs to differentiate into podocytes [18]. To this aim, cells were seeded on different substrates and induced to differentiate toward the podocyte lineage by 48 h incubation with all-trans retinoic acid (RA) [18]. At the end of the incubation period, differentiation was evaluated assessing the mRNA expression levels of several podocyte-specific markers by qRT-PCR and by analysis of expression levels of the podocyte-specific protein nephrin by immunofluorescence. When RPCs were plated on stiff substrate, their differentiation toward podocytes increased in comparison to cells cultured on soft substrates, as demonstrated by the higher expression of Klf15 (Figure 5d) and nephrin at both mRNA (Figure 5e) and protein level (Figure 5g,h). The expression of synaptopodin increased with stiffness but the changes were not significant (Figure 5f). Interestingly, the expression of both Klf15 and nephrin was found to be higher in cells cultured on substrates having a stiffness of around 4–12 kPa than on both the softer and the harder substrates (Figure 5d,e,g,h).

These results demonstrate that the capacity of RPCs to differentiate into podocytes is modulated by the elastic modulus of the substrate and that stiffness variations over a limited range strongly influence the cellular response.

### 3.5. ROCK Activity is Required to Mediate the Effects of Stiffness on RPC Proliferation, Migration and Differentiation

The activation of ROCK is known to modulate the organization of the actin-based cytoskeletal systems, including the formation of stress fibers and FA. In order to understand the role of ROCK in RPC response to extracellular stiffness, we treated RPCs grown on stiff substrates (12 and 50 kPa) with the ROCK inhibitor Y27632, analyzed actin cytoskeleton organization, and paxillin distribution. Inhibition of ROCK resulted in the disassembly of stress fibers and FA, as evidenced by F-actin disorganization and reduction of paxillin clusters (Figure 6a–c).

We next explored the involvement of ROCK activity on the stiffness-mediated increase in RPC proliferation and migration. In the MTT assay, we observed that treatment with Y27632 inhibited the proliferation of RPCs cultured on substrates with higher stiffness, while no effects were observed on RPCs cultured on soft substrates (Figure 6d), indicating that ROCK activity is required to mediate the effect of stiffness on RPC proliferation.

Similar results were obtained analyzing the effect of Y27632 treatment on RPC migratory capacity. Indeed, RPCs cultured on a 50 kPa substrate in presence of Y27632 showed a reduction in their overall migration efficiency, as demonstrated by the significant decrease observed in MSD (Figure 6e).

Treatment of RPCs with Y27632 during the differentiation period inhibited the increase of nephrin expression induced by RA of around 80% (Figure 6f), while the increased expression of the transcription factor Klf15 was not altered by the presence of the ROCK inhibitor (Figure 6g). Interestingly, nephrin is a component of the slit diaphragm, that together with focal adhesions constitute a molecular complex that in podocytes functions as primary signaling hub that maintains a dynamic control of the actin cytoskeleton. These results underline the primary role of nephrin in orchestrating the signaling network between the slit diaphragm and the actin cytoskeleton and suggest a primary role for ROCK activation on slit diaphragm organization.

### 3.6. ROCK1 and ROCK2 Mediate the Effects of Stiffness on RPC Proliferation, and Migration

Y27632 is a non-isoform-selective pharmacological inhibitor of both ROCK1 and ROCK2 and also has non-selective effects. To more selectively target ROCK1 and ROCK2 isoforms and verify their roles on the stiffness mediated effects on RPC physiology, we treated RPCs with antisense-LNA-GapmeRs against ROCK1 or ROCK2. LNA-GapmeRs induced downreguation of ROCK1 and ROCK2 mRNA and protein expression (Figure 7a–c). Moreover, immunofluorescence for ROCK1 and ROCK2 demonstrated upregulation of both isoforms on RPCs cultured on stiff (50 kPa) substrates in comparison to cells cultured on soft (0.5 kPa) substrate (Figure 7b,c) and that LNA-GapmeRs strongly reduced protein expression of both isoforms (Figure 7b,c). Inhibition of both ROCK1 and ROCK2 resulted in the disassembly of FA in RPCs seeded on stiff substrates, as evidenced by reduction of paxillin clusters (Figure 7d). In agreement with the results obtained with Y27632, downregulation of both ROCK1 and ROCK2 with LNA-GapmeRs inhibited the proliferation of RPCs cultured on substrates with higher stiffness in comparison to soft substrates (Figure 7e), and their migration efficiency, as demonstrated by the significant decrease observed in both MSD and speed (Figure 7f,g).

## 4. Discussion

Discovering the influence of matrix stiffness on SC behavior has important implication for SC biology and RM. The effects of matrix stiffness have been largely studied for ESCs, MSCs, neural SCs, corneal epithelial SCs, and muscle SCs. In this study, the use of collagen I-coated polyacrylamide hydrogels with variable stiffness indicated that, at least in vitro, the phenotype of human RPCs was highly dependent on the Young’s modulus of the substrate, with stiffer substrates promoting RPC proliferation and migration. Substrate stiffness modulated also the capacity of RPCs to differentiate toward podocytes, with a Young’s modulus of 12 kPa being optimal among those analyzed.

Our results demonstrated that actin cytoskeleton reorganization with formation of actin stress fibers, FA assembly and ROCK activation represent important molecular elements underlying the mechanotransduction responses of human RPCs. There are many mechanisms at play at the cell/substrate interface, but the fundamental interaction that all cells must have is a link between the cytoskeleton and the substrate [45]. Following this interaction, a cascade of events in the cells starts, all of which are initiated by the cytoskeleton. One of these mechanisms is the generation of contractile forces by cells against the substrate [46]. To contract, cells must bind to the protein grafted to the substrate though integrins that are part of the FA that bind to actin [46]. The Rho family of GTPases consists of more than 20 proteins that regulate specific effects of the actin cytoskeleton [47]. Their activity correlated with material stiffness leading, in MSCs, to spreading and shape modification, migration and differentiation into contractile lineages [26]. Similarly, RPCs seeded on stiff substrates were typically well-spread with brighter F-actin displaying a bundle-like distribution and paxillin clusters localized at the end of bundle-like actin microfilaments. Treatment with the ROCK inhibitor Y27632 disrupted this cytoskeleton organization resulting in lower capacity to proliferate, migrate and differentiate into podocytes in response to the stiffening of the substrate. Y27632 is a non-isoform specific pharmacologic inhibitor of both ROCK1 and ROCK2 isoforms and also has non-specific effects [47]. Thus, we confirmed the role of ROCK isoforms by performing the same experiments using RPCs treated with LNA-GapmeRs from ROCK1 and ROCK2, with similar results.

To our knowledge, this is the first paper focused on the effects of substrate mechanical properties on RPCs. Few studies have analyzed these effects on other renal cell types. Beamish et al. studied the in vitro effects of substrate mechanical properties on human renal proximal tubular cells (RPTEC) [48]. In agreement to our data, they observed that increased substrate stiffness promotes RPTEC spreading, proliferation, and FA activation, effects in part related to activation of extracellular signal-regulated kinase (ERK).

Hu et al. [49] and, more recently, Abdallah et al. [31] studied the effects of substrate mechanical characteristics on podocytes. Hu et al. reported a pro-differentiation and maturation effect of gelatin substrates with Young’s modulus near that of healthy glomeruli (in the 2–5 kPa range) and suggested that Src-mediated activation of RacGAPs may be responsible for the mechanoresponse of podocytes [31]. Abdallah et al. obtained similar results using hydrolyzed polyacrylamide hydrogel as substrate for podocyte culture. On this substrate, podocytes spread, formed a dense actin cytoskeleton and proliferated more as the Young’s modulus of the gel increased [31]. Moreover, in agreement with Hu et al., they observed that podocin level of expression was higher in cells cultured on substrates having an elasticity within the range of healthy glomeruli [31].

With this study, we demonstrated that substrate stiffness induced dramatic effects on RPC shape and activities. This knowledge could increase our ability to control RPC behavior at least in vitro using not only biochemical supplements, but also inherent material properties and thus help to circumvent cost barriers to efficient RPC expansion and lineage-specific differentiation. Moreover, the results described in this work could have important implications for the design of new renal assist devices, in which the surfaces could be coated or physically modified to encourage cell integration and promoting appropriate RPC responses. Finally, the comprehension of the physical forces that regulates RPC behavior are crucial for the development and optimization of glomerular-on-chip models. In this context, it is of interest that polyacrylamide hydrogels can be designed to simulate the in vivo environment and should be regarded as a promising scaffold material.

RM strategies based on activation of endogenous regeneration operated by RPCs would also benefit from the enhanced understanding of their response to extracellular biophysical properties provided in this work. Indeed, in vivo mechanical cues are important in adult tissue homeostasis, where adult SCs require physical interactions with the ECM to maintain their potency and the alteration of tissue stiffness is associated with diverse array of pathologies [26]. Tissues stiffen during aging and during the pathological progression of cancer and fibrosis [26]. Moreover, ECM stiffness is emerging as a prominent mechanical cue that precedes disease and drives its progression by altering cellular behaviors [26]. Targeting ECM mechanics, by preventing or reversing tissue stiffening or interrupting the cellular response, is a therapeutic approach with clinical potential [50]. For example, Gouveia et al. [51] showed that substrate stiffness within the native limbal epithelial stem cell (LESC) niche is relevant to SC phenotype and wound healing and that treating the fibrotic, burnt surface of the cornea with collagenase restored the mechanical properties of the tissue and its capacity to support LESCs [51].

Little is known about the biomechanical properties of healthy and diseased kidneys. Using atomic force microscopy, Wyss and colleagues found that the Young’s modulus of healthy rat and mouse glomeruli is ~2.5 kPa, and demonstrated that at an early stage of injury in Col4a3(-/-) mice (Alport model) and Tg26^HIV/nl^ mice (HIV-associated nephropathy model), the glomeruli were significantly softer than normal [52]. The overall reduction in the Young’s modulus of glomeruli was around 30% and came before the appearance of glomerular structural damage (e.g., glomerular sclerosis) [52]. This was an unexpected result, as one would predict glomeruli at early stages of disease to be normal and then stiffen with progressive sclerosis. Similarly, Embry et al. [27] reported that in Tg26^HIV/nl^ mice, glomeruli soften progressively despite an increased collagen deposition. This observation could be in agreement with our demonstration of increased mRNA levels of ECM components in RPCs cultured on softer substrates.

Our results have potential relevance to understanding the mechanisms of glomerular regeneration operated by RPCs. We can hypothesize that, because of the softening of glomeruli in the early stage of glomerular injury, RPCs assume a more undifferentiated phenotype and start the deposition of matrix components. Increased matrix deposition induces increased stiffness of the tissue that may permit maintenance of structural integrity at the cost of function but allow for subsequent recovery, inducing RPC activation, and increasing RPC proliferation and migration. Indeed, the acquisition of a more migratory phenotype, allowed by the promotion of FA in stiff conditions, is an important feature in wound healing responses. Interestingly, most cells migrate preferentially to stiffer regions via unbalanced forces created by stiffness gradients, i.e., durotaxis. The adhesion of parietal epithelial cells to bare glomerular filtration membrane observed in the early stage of glomerular diseases following podocyte detachment could represent the mechanism by which RPCs localized on the parietal layer of the Bowman’s capsule “sense” the glomerular tuft stiffness. From our data, it is also possible to hypothesize that RPCs signal changes in biophysical cues to other cells within the glomerulus trough thin cytoplasmic bridges. Indeed, we observed in vitro the presence of long protrusions resembling nanotubes on RPCs grown on soft substrates similar to those observed in vivo in a mouse model of unilateral ureteral obstruction using multiphoton microscopy [53].

Interestingly, enhanced expression of paxillin and FA kinase have been reported in glomeruli of rat models of glomerular immune injury and in experimentally induced nephrotic syndrome [54,55], while modulation of paxillin activity and FA formation appear as a putative therapeutic target for podocytopathies [56]. Taken together, these data suggest that reaching and maintaining an appropriate level of glomerular stiffness contributes to retain a healthy glomerulus, and this happens, at least in part, by exposing RPCs to the ideal biomechanical condition to support their proliferation, migration and differentiation toward podocytes.

## 5. Future Directions

Creating an optimized microenvironment suitable to support growth and differentiation of regenerative SCs is becoming an area of intense research, in particular for biomaterial-mediated tissue regeneration. Indeed, with the advancement of material science, it is becoming possible to generate active biophysical signals to direct SC fate through specially designed material microstructures. These biophysical signals include material stiffness, tensile strength, micropatterned structure, material surface chemistry, ECM-coated material, and material-transmitted extracellular mechanical force. Future researches could unveil how these material properties can locally control the expansion and direct the differentiation of specific endogenous steSC populations, or serve as inductive regulators for de novo tissue formation, with important implications for in situ tissue engineering approaches.

## Figures and Tables

**Figure 1 cells-08-01561-f001:**
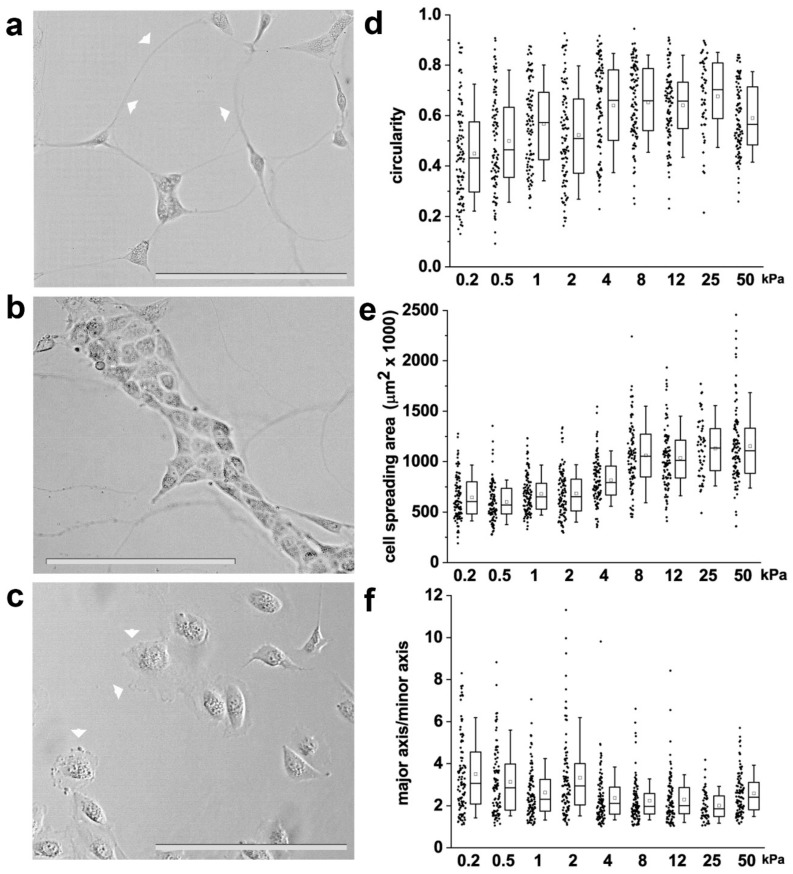
Substrate stiffness affects renal progenitor cell (RPC) morphology in vitro. (**a**,**b**) Representative optical phase contrast images of RPCs grown on substrates of 0.5 kPa Young’s modulus. Arrowheads point to long protrusions resembling tunneling nanotubes. (**c**) Representative optical phase contrast images of RPCs grown on substrate of 50 kPa Young’s modulus. Arrowheads point to sheet-like membrane protrusions resembling lamellipodia. (**d**) Quantification of single cell circularity, (**e**) cell area and (**f**) major to minor axis ratio. At least 70 cells were analyzed from two independent experiments. Box-and-whisker plots: line = median, box = 25–75%, whiskers = 10–90%. Statistical analysis was performed using ANOVA with Tukey post-hoc test. For statistical analysis results, see Appendix A. Bars = 200 µm.

**Figure 2 cells-08-01561-f002:**
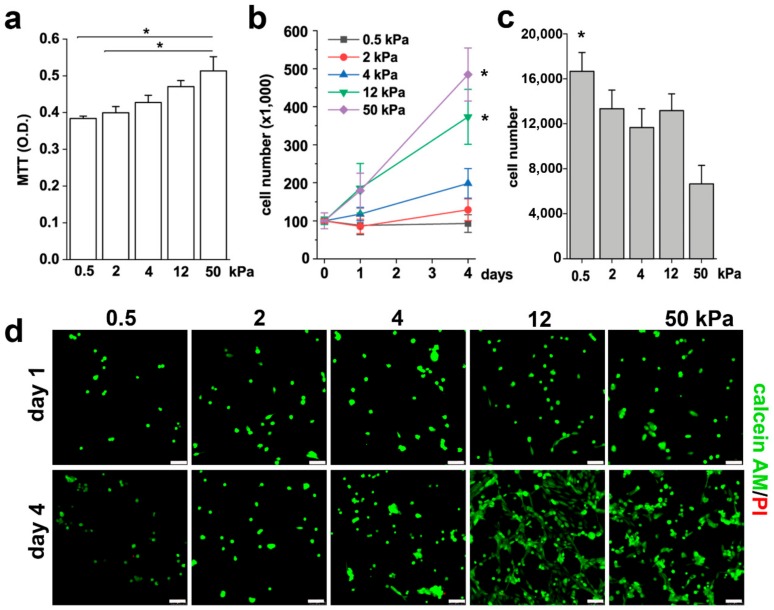
Substrate stiffness modulates RPC proliferative capacity in vitro. (**a**) MTT assay of RPCs performed 48 h after plating on substrates with different stiffness. Data are expressed as mean ± SE. (**b**) Growth curves obtained culturing RPCs for 4 days on substrates of different stiffness. Data are expressed as mean ± SE (replicate n = 6). (**c**) Number of RPCs present in the surnatant 24 h after plating. Data are expressed as mean ± SE (replicate n = 2). (**d**) Confocal images of Calcein AM (green fluorescence) and propidium iodide (red) staining in RPCs at day 1 and day 4 after plating. * *p* < 0.05, using one-way ANOVA with Tukey post-hoc test. Bars = 75 µm.

**Figure 3 cells-08-01561-f003:**
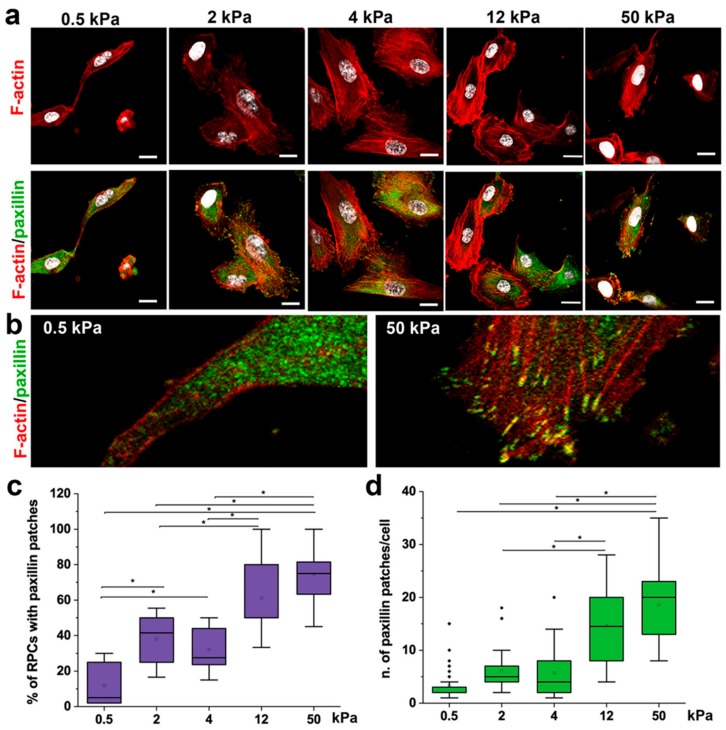
Substrate stiffness modulates cytoskeleton organization and FA formation. (**a**) Confocal images of F-actin immunodetection by phalloidin (red), paxillin (green) and nuclei with DAPI counterstain (white) of RPCs cultured on substrates with different stiffness. F-actin organization shows a trend, from diffuse on soft gels to progressively organized on stiffer substrates (as stress fibers). (**b**) Higher magnification images showing that paxillin staining was diffuse on soft substrate (left), or organized in clusters on the cell membrane in stiff conditions (right). (**c**) Percentage of RPCs containing paxillin clusters in function of stiffness. At least 10 representative images from each condition were analyzed. (**d**) Average number of paxillin patches in cell cultured on different stiffness. At least 20 cells for each condition were analyzed. Box-and-whisker plots: line = median, box = 25–75%, whiskers = 10–90%. **p* < 0.05 using one-way ANOVA followed by Tukey’s post-hoc test. Bars = 25 µm.

**Figure 4 cells-08-01561-f004:**
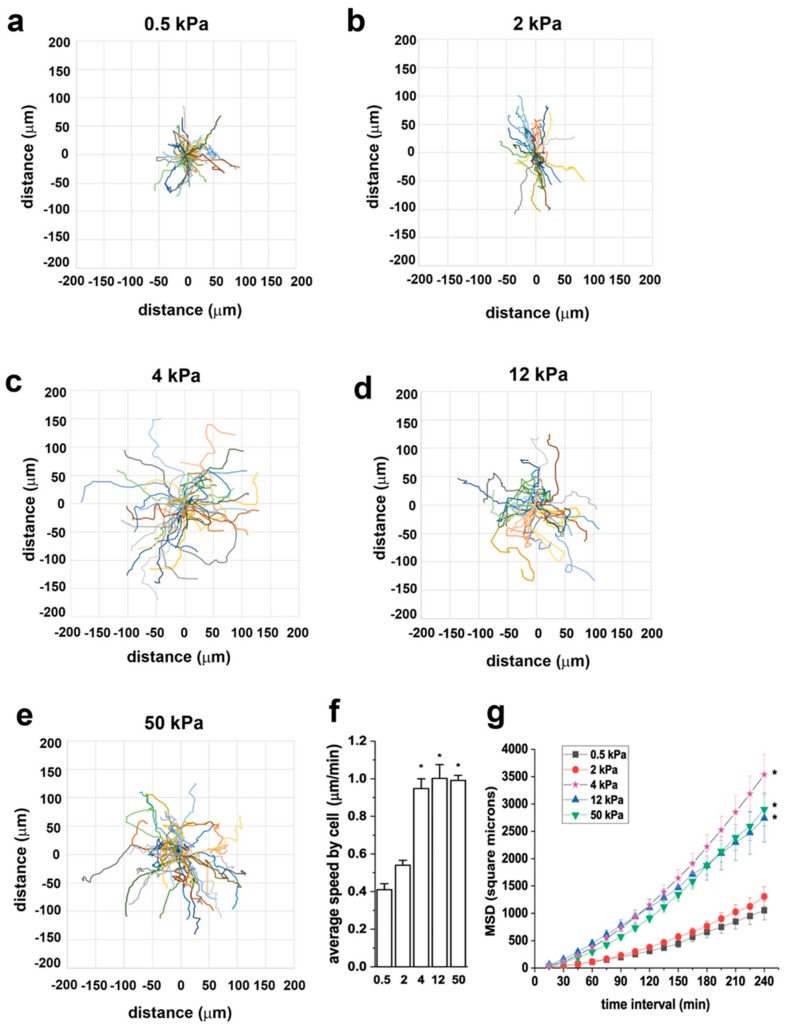
Substrate stiffness modulates RPC migratory capacity in vitro. (**a**–**e**) Wind rose plots of cell trajectories on 0.5, 2, 4, 12, and 50 kPa. At least 30 randomly selected cell trajectories over 3 h are shown for each condition. (**f**) Average speed of RPCs cultured on substrates with different stiffness. At least thirty randomly selected RPCs were monitored over 3 h for each condition. (**g**) Mean square displacement (MSD) of RPCs cultured on substrates with different stiffness (n = 90, 36, 45, 32, 46 trajectories for 0.5, 2, 4, 12, and 50, respectively). Data are expressed as mean ± SE. **p* < 0.05 using one-way ANOVA followed by Tukey’s or Bonferroni post-hoc test, as appropriate.

**Figure 5 cells-08-01561-f005:**
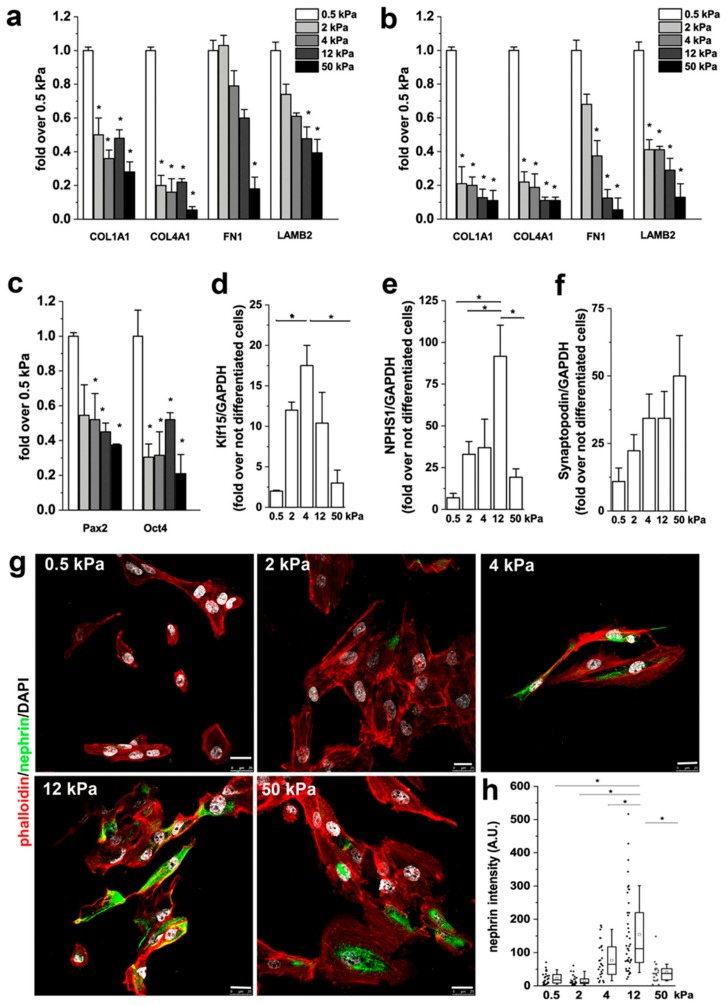
Substrate stiffness controls podocyte differentiation of RPCs. (**a**) mRNA levels of extracellular matrix (ECM) components as assessed by qRT-PCR in RPCs. (**b**) mRNA levels of ECM components assessed by qRT-PCR in RPCs following 48 h treatment with 10 ng/mL transforming growth factor (TGF)-β. (**c**) mRNA expression of Pax2 and Oct-4 after 2 weeks in culture. (**d**–**f**) mRNA levels for Klf15, nephrin (NPHS1) and synaptopodin. Data are mean ± SE of two independent experiments (replicate n = 3). (**g**) Confocal images of F-actin, immunodetected by phalloidin (red), nephrin (green) and nuclei with DAPI counterstain (white), in RPCs differentiated toward podocytes on substrates with different stiffness. (**h**) Quantification of nephrin fluorescence intensity in RPCs as a function of stiffness. At least 20 cells for each condition were analyzed. Box-and-whisker plots: line = median, box = 25–75%, whiskers = 10–90%. Scale bar is 25 μm. **p* < 0.05 using one-way ANOVA followed by Tukey’s test.

**Figure 6 cells-08-01561-f006:**
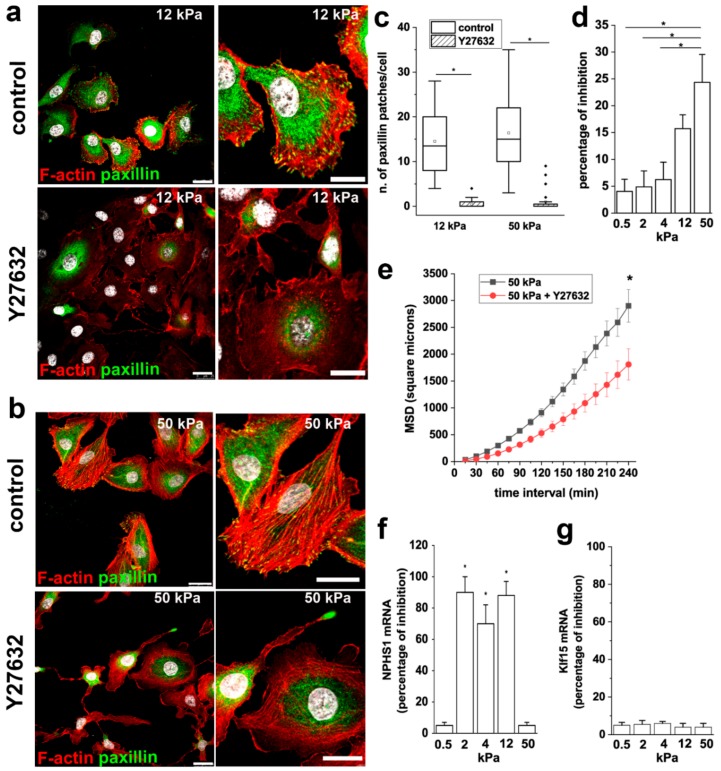
ROCK activity is required to mediate the effects of stiffness on RPC proliferation, migration and differentiation. (**a**,**b**) Confocal images of F-actin immunodetection by phalloidin (red), paxillin (green) and nuclei with DAPI counterstain (white) in RPCs cultured on (**a**) 12 kPa and (**b**) 50 kPa substrate treated with vehicle (DMSO, control) and with the Rho kinase inhibitor Y27632. Bars = 25 μm (**c**) Average number of paxillin patches in RPCs cultured on 12 and 50 kPa substrates in control condition and treated with Y27632. At least 20 cells for each condition were analyzed. Box-and-whisker plots: Line = median, box = 25–75%, whiskers = 10–90%. (**d**) Percentage of growth inhibition induced by 10 μM Y27632 treatment of RPCs assessed by MTT assay. Data are mean ± SE (replicate n = 8). (**e**) MSD of RPCs cultured on 50 kPa stiffness in presence or absence of Y27632 (n = 46 and 44 trajectories for 50 kPa and 50 kPa + Y27632, respectively). (**f**) Percentage of inhibition of nephrin expression induced by 10 μM Y27632 treatment. (**g**) Percentage of inhibition of Klf15 expression induced by 10 μM Y27632 treatment. Data are expressed as mean ± SE. **p* < 0.05 using one-way ANOVA followed by Tukey’s or Bonferroni post-hoc test, as appropriate.

**Figure 7 cells-08-01561-f007:**
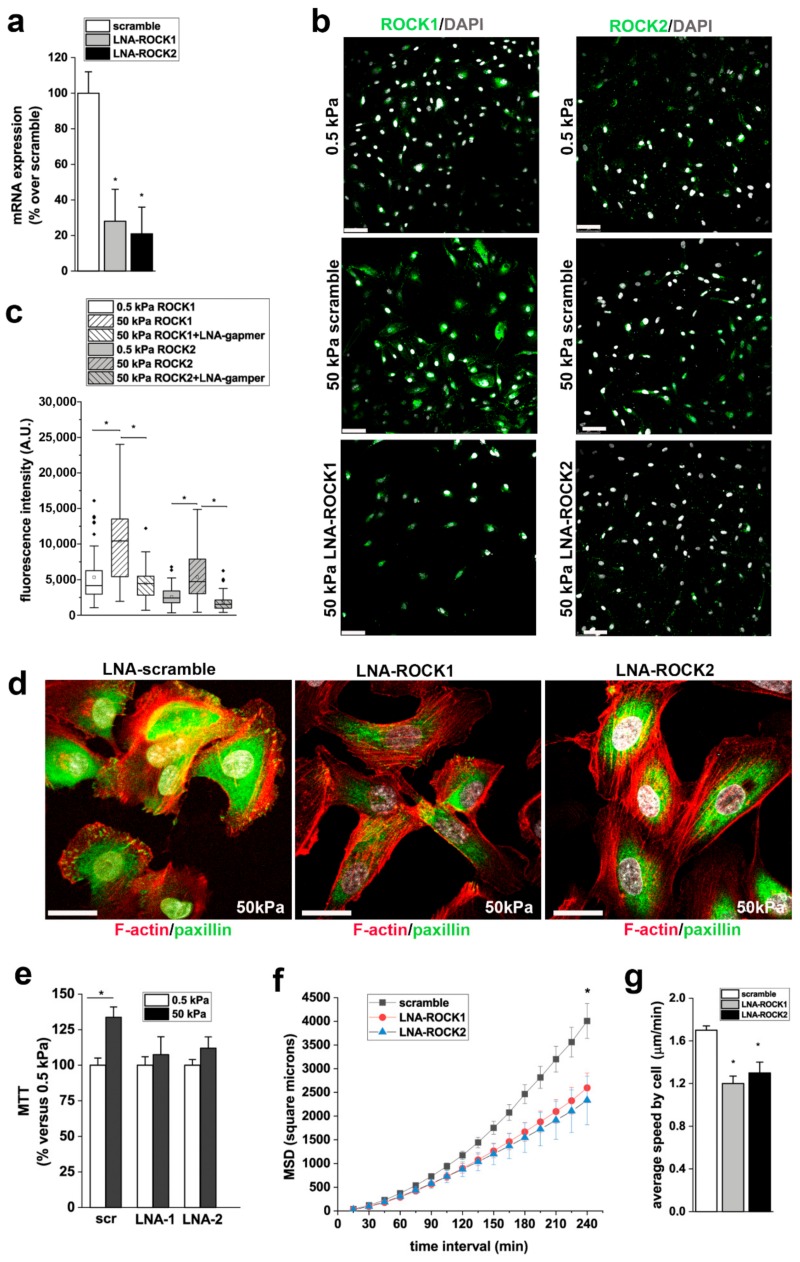
ROCK 1 and ROCK2 mediate the effects of stiffness on RPC proliferation and migration. (**a**) ROCK1 and ROCK2 mRNA expression in scramble and LNA-GapmeRs treated RPCs, as assessed by RT-PCR (n = 2 experiments). (**b**) ROCK1 (top) and ROCK2 (bottom) expression (green) in RPCs cultured on 0.5 kPa substrate (left) and on 50 kPa substrates 48 h after transfection with scramble and ROCK1 and ROCK2 LNA-GapmeRs. DAPI counterstains nuclei (white). Bars = 75 μm. (**c**) Quantification of ROCK1 and ROCK2 fluorescence intensity in RPCs cultured on substrate with 0.5 kPa stiffness and in RPCs cultured on 50 kPa substrates 48 h after transfection with scramble and ROCK1 and ROCK2 LNA-GapmeRs. At least 50 cells for each condition were analyzed. Box-and-whisker plots: line = median, box = 25–75%, whiskers = 10–90%. (**d**) Confocal images of F-actin immunodetection by phalloidin (red), paxillin (green) and nuclei with DAPI counterstain (white) in RPCs cultured on 50 kPa substrate treated with scramble and with ROCK1 and ROCK2 LNA-GapmeRs. Bars = 25 μm. (**e**) Percentage of cell viability in scramble and LNA-ROCK1 and ROCK2 treated RPCs cultured on 50 kPa substrate versus RPCs grown on 0.5 kPa substrate. Data are mean ± SE (replicate n = 4). (**f**) MSD of RPCs treated with scramble or LNA-GapmeR for ROCK1 and ROCK2 and cultured on 50 kPa stiffness (n = 45, 30, and 32 trajectories for 50 kPa + scramble, 50 kPa + LNA-ROCK1, and LNA-ROCK2, respectively). (**g**) Average speed of RPCs transfected with scramble or LNA-GapmeR for ROCK1 and ROCK2 and cultured on 50 kPa. At least thirty randomly selected RPCs were monitored over 3 h for each condition. Data are expressed as mean ± SE. **p* < 0.05 using one-way ANOVA followed by Tukey’s or Bonferroni post-hoc test, as appropriate.

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
