# Peer review of "Substrate Stiffness Modulates Renal Progenitor Cell Properties via a ROCK-Mediated Mechanotransduction Mechanism"

_cells, 2019, doi:10.3390/cells8121561_

Round 1

Reviewer 1 Report

Melicia et al., show that the stiffness of the matrix induces focal adhesion assembly trough Rho kinase mediated mechanisms, and this affects cell spreading as well as proliferative and migratory capacity of the cells. The implications of these findings could improve renal stem- and progenitor cell proliferation in vitro and in vivo applications.

This is an excellent study and experiments are well described and characterized. I would recommend the following experiments to further strengthen the paper:

long term culture of the cells (influenced by the substrate stiffness). siRNA to knock out ROCK and corroborate the Y27632 experiments.

In some instances it is mentioned 'data not shown' - kindly include these.

Reviewer 2 Report

I commend the authors for the good work conducted throughout. However, the research novelty and innovation could have been improved since other authors are embarking on similar trends. Regardless, the authors have demonstrated that their results suggest that the capacity of RPCs to differentiate into podocytes can be modulated by the elastic modulus of the substrate.  

Suggestions – In the discussion, the authors can elaborate further by mentioning a section towards ‘future directions’, where material surface chemistry may influence the effect of the ECM remodelling and tissue micro-environment particularly, in tissue regeneration responses. This approach has not been mentioned and will enhance the interest of readers and development in the area of research. Furthermore, material tensile strength testing can be mentioned. Collectively, the work will contribute towards the enhancement of the knowledge in the field.

Reviewer 3 Report

The authors claim that substrate stiffness modulates renal progenitor cell properties via a ROCK-mediated mechanotransduction mechanism as the title described. Before publication, the points, mentioned below, need to be clarified or amended.

In Figure 6, if the effect of stiff matrix are disappeared by ROCKi, presenting the change of Paxillin in F-actin12 or 50 kPa groups and 12 or 50 kpa + ROCKi will be recommended.. The title is too definitive. If the authors to keep their title, protein level of analyses such as Western blotting or IF of ROCK1, ROCK2should be presented. Figure 1d-f need marks of significances. In Figure 2, are the cells detached or apoptic? In Figure 2c, to avoid the possibility of the difference of the number of initial seeded cells, additional experiments or explanation is required Page 12, Line 301: ‘TGF-β) mRNA expression’ may be changed as ‘TGF-β. (c) mRNA expression’

Round 2

Reviewer 1 Report

Authors have addressed the concerns